# Something Borrowed, Something New: A Governance and Social Construction Framework to Investigate Power Relations and Responses of Diverse Stakeholders to Policies Addressing Antimicrobial Resistance

**DOI:** 10.3390/antibiotics8010003

**Published:** 2018-12-24

**Authors:** Helena Legido-Quigley, Mishal S. Khan, Anna Durrance-Bagale, Johanna Hanefeld

**Affiliations:** 1London School of Hygiene and Tropical Medicine, London WC1H 9SH, UK; mishal.khan@lshtm.ac.uk (M.S.K.); anna.durrance-bagale@lshtm.ac.uk (A.D.-B.); johanna.hanefeld@lshtm.ac.uk (J.H.); 2Saw Swee Hock School of Public Health, National University of Singapore, 119077 Singapore, Singapore

**Keywords:** Antimicrobial resistance, policy process, social construction, governance, power relations

## Abstract

While antimicrobial resistance (AMR) has rapidly ascended the political agenda in numerous high-income countries, developing effective and sustainable policy responses in low- and middle-income countries (LMIC) is far from straightforward, as AMR could be described as a classic ‘wicked problem’. Effective policy responses to combat AMR in LMIC will require a deeper knowledge of the policy process and its actors at all levels—global, regional and national—and their motivations for supporting or opposing policies to combat AMR. The influence of personal interests and connections between for-profit organisations—such as pharmaceutical companies and food producers—and policy actors in these settings is complex and very rarely addressed. In this paper, the authors describe the role of policy analysis focusing on social constructions, governance and power relations in soliciting a better understanding of support and opposition by key stakeholders for alternative AMR mitigation policies. Owing to the lack of conceptual frameworks on the policy process addressing AMR, we propose an approach to researching policy processes relating to AMR currently tested through our empirical programme of research in Cambodia, Pakistan, Indonesia and Tanzania. This new conceptualisation is based on theories of governance and a social construction framework and describes how the framework is being operationalised in several settings.

## 1. Antimicrobial Resistance—A ‘Wicked Problem’ Requiring an Understanding of Process, Actors and Power

Owing largely to its recognition as a major threat to global health security, AMR has rapidly ascended the political agenda in numerous high-income countries (HICs). It has also been noted that AMR could jeopardise efforts to achieve Sustainable Development Goal (SDG) 3 (good health and wellbeing) and could also adversely impact progress toward other SDGs since AMR predominantly impacts the poor, threatens sustainable food production, and can contribute toward water contamination [1]. As a result, substantial funding—with estimates in the region of 40 billion USD over the next ten years—is being sought to tackle AMR, and the World Health Assembly endorsed a Global Action Plan on AMR in May 2015 [2]. The Global Action Plan on AMR adopts a One Health approach and relies on three international policy bodies—the World Health Organization (WHO), the Food and Agricultural Organisation (FAO) and the World Organisation for Animal Health (OIE)—to facilitate the preparation of national AMR action plans, which all WHO member states have been urged to develop by 2017. Despite the concern and momentum on AMR in HICs, developing effective and sustainable policy responses in low- and middle-income countries (LMICs) will be far from straightforward, as AMR could be described as a classic ‘wicked problem’. Wicked problems have two key characteristics, clearly represented in the AMR challenge [3]. First, they involve a great number of stakeholders, often with conflicting interests and complex networks. This is the case with the development of AMR policy responses, since the emergence and spread of AMR is driven by activities in the areas of human and animal health, food production and interactions with the environment, thus requiring a ‘One Health’ approach [4] built upon collaboration between sectors. Furthermore, the commercial sector including for-profit organisations such as pharmaceutical companies and the food industry have an interest in many of the aspects of AMR control, including policies and regulation promoting antimicrobial stewardship. Therefore, the diversity of stakeholders involved in comprehensive AMR responses is particularly high. 

The second key feature of wicked problems is the critical role of problem framing in driving a response. In HICs, AMR has been framed in terms of the catastrophic health and economic impact it would have, with a clear link to security concerns and a ‘fear factor’ in the public discourse surrounding AMR. This is illustrated by a statement on the UK Review on AMR (2014) made by UK Prime Minister David Cameron: If we fail to act, we are looking at an almost unthinkable scenario where antibiotics no longer work and we are cast back into the dark ages of medicine [5]. While this approach has been fairly successful in directing the attention of publics and policymakers toward AMR in HIC contexts, it is unclear how AMR should be framed to generate an effective policy response in LMICs. 

There is a dearth of empirical evidence available to guide framing of AMR to generate sustained political attention and policy responses in LMICs, or indeed to identify which stakeholders should be engaged in the policy process. However, policy analysis for health has often examined challenges of intersectoral working and policy-processes spanning multiple sectors [6], such as trade and health [7]. Thus, conceptual approaches and methods for AMR policy analysis may be adapted from other areas. Here we describe a novel conceptual framework to examine policy processes, social constructions and power relations—including actors which are not traditionally considered at the centre of global health issues—and how the phrasing of a health issue affects the policy process.

## 2. Methods: Literature Reviewed to Develop Conceptual Framework

To develop our conceptual framework, we relied on seminal papers in three areas: governance, social construction and the policy process in LMICs (8–21). Two of the authors reviewed this literature and identified key concepts on governance and how AMR and target populations of potential interventions are socially constructed, while the other two authors operationalised the conceptual framework into a methodology to apply in a range of countries. In the sections that follow, we summarise how the literature we reviewed shaped our conceptual framework and describe the processes used to operationalise the framework in a range of countries.

## 3. A Proposed Conceptual Framework for AMR Policy Analysis

We present here a conceptual framework for policy analysis on AMR. Our approach is stepwise, focusing on the overall context and processes first, then, moving to governance, before highlighting how social construction of an issue interplays with context and governance and determines who is seen as critical in addressing AMR. The next paragraphs describe each of the constructs in detail including its associated theoretical underpinnings. The importance of power relations is a cross-cutting theme present at all levels of analysis. 

### 3.1. Policy Context: Features of the Policymaking Environment that Can Influence Policy Decisions

Our proposed framework is based on theories of the policy process and political economy as well as reflecting on some key sociological concepts such as power relations and governance. As Reich (1995) and Walt and Gilson (1994) argued, policy reform is fundamentally a political process, which affects the formulation and implementation of policy [8,9]. The seminal work of Walt and Gilson (1994) argued that health policy focuses too much attention on the content of the policy, while neglecting the actors, the processes involved in implementing change and the context in which the policy originates [9]. As a result of this work and that of other colleagues, the field of health policy has evolved to include more explicit attention on processes [10]. Reasons for inconsistent policy implementation or indeed failure, are frequently attributed to the adaptation of inappropriate ‘technical solutions’ or missing inputs or products. Instead, the outcomes of a new policy are often a result of a complex interplay of dynamic factors, including the context in which a policy is introduced and implemented, its content and the process through which it is introduced with policy actors, and their relationships at the centre of this interplay [11]. The AMR policy context represents a policy arena where politics and conflict are more evident than in many other areas of health policy. Diverse professional groups—representing human and veterinary medicine, agriculture, the environment, pharmaceuticals and other economic interests—dominate the AMR policy arena. These bring often competing interests, and strong ideas of what policy options ought to be adopted. In addition, there are also multiple political obstacles to addressing AMR, as policy decisions can affect profits of various groups (pharmaceutical industry, food producers, for-profit healthcare providers) and have ramifications for government ministries beyond health, such as trade, tourism and agriculture.

### 3.2. Governance Framework: Actors, Institutions, Power Relations and Coalitions

Given this context we find governance related issues play a large role in AMR but are poorly studied. We draw on the definition of governance provided by the Commission on Global Governance [12] as: “the sum of the many ways individuals and institutions, public and private, manage their common affairs. It is a continuing process through which conflicting or diverse interests may be accommodated and cooperative action may be taken. It includes formal institutions and regimes empowered to enforce compliance, as well as informal arrangements that people and institutions either have agreed to or perceive to be in their interest.”

There are two main acceptations of governance relevant to AMR. One explores how overall governance of a system or indeed ‘the whole-of-society’ affects an issue, such as AMR. The second one focuses on specific mechanisms governing a specific field, such as the health, animal, food production and environmental sectors. First, there is a need to examine a broader conceptualisation of governance, which is often called “governance for health”. This is defined as “the attempts of governments or other actors to steer communities, countries or groups of countries in the pursuit of health as integral to well-being through both whole-of-government and whole-of-society approaches” [13]. As Illona Kickbusch acknowledges, in recent years we have seen a diffusion of governance, which also applies to AMR policy, moving from a model dominated by the state to a new model where governance is co-produced by a wide range of actors present at local, national and global level [13]. Within this new policy scenario, Kickbusch has identified five types of governance present in whole-of-government approaches which can also be reflected upon when studying AMR policy process. These include: governing by collaborating; by engaging citizens; by mixing regulation and persuasion; through independent agencies and expert bodies; and by adaptive policies, resilient structures and foresight.

A second level of governance is what has been called “health governance” referring to the governance of the health system which includes "the processes, structures and organizational traditions that determine how power is exercised, how stakeholders have their say, how decisions are taken and how decision-makers are held to account” [13]. In studying AMR governance, “health governance” needs to be expanded to include the processes and stakeholders represented in the animal, food production and environment sector, moving from an analysis of “health governance” to one prioritising “One Health”. 

Considering governance approaches for the successful control of AMR requires defining specific dimensions to assess how governance is exercised, these include among others: shared vision, multilevel governance coordination, priority setting, reported regulations, performance monitoring, differing power relations between different professional groups, and existing accountability mechanisms [14]. It requires a focus on both whole of society and sector specific approaches.

### 3.3. Identify How AMR Is Being Socially Constructed

Next, following on from considerations of governance we suggest a focus on the social construction of AMR. As Wernli et al (2017) have suggested and as mentioned earlier, there are different ways in which AMR has been socially constructed [15]. The authors have identified five different frames, with a distinguished set of actors and corresponding policies for each of the frames. The first frame, a One Health approach, combines in one paradigm human and animal health as well as taking into consideration the environment which has been more recently identified as key in addressing the AMR response. The second frame considers AMR as a health security threat and has been assimilated into the Global Health Security agenda. The third frame explores AMR as a healthcare policy issue where AMR is seen as a biomedical problem within the context of healthcare facilities and within the pursuit of clinicians controlling and preventing infectious diseases. The fourth frame considers AMR as a development issue following the premise that LMICs have a higher burden of infectious diseases and are affected by lack of sanitation, poor hygiene, and misuse of antibiotics. Finally, the fifth frame sees AMR as an innovation issue highlighting the need for research and development of new antibiotics and diagnostics [15]. Wernli et al (2017) also distinguish a set of actors and policies for each of the frames described. For example, through a One Health frame, a collaboration has been fostered between WHO, FAO and OIE and policies have translated into the WHO Global Action Plan with most countries incorporating some of the One Health ideas into their National Action Plans [15]. 

However, while these frames and corresponding policies are a useful way to analyse AMR policy in a particular setting, there is one crucial component missing in this approach. There is a need to incorporate into the analysis who the target populations are and how the different actors are influenced by AMR policies. So not only who constructs AMR as a frame, but also who is targeted by policies and interventions seeking to respond to AMR. The selected frames and how the target populations are socially constructed are essential to understand the processes and the “how” of policy design and implementation. For example, in on-going research in Pakistan we identified a framing of AMR very strongly as a threat to human health, while control measures evident in current documents focused almost exclusively on livestock and farming. We argue that the framing of the issues relating to AMR, together with the selection of policy options to address AMR, and the population or actors they target themselves represent an exercise of power. To understand these power relations is essential if we want to ensure successful implementation of policies designed to respond to the threat of resistance. 

### 3.4. Adopting the Social Construction Framework towards Target Populations

To understand how policy options identify and affect target populations we propose to adopt the Social Construction Framework (SCF) for Public Policy [16]. The SCF highlights how policies are created in an environment of uncertainty, where it is believed that individuals make decisions subjectively, mostly prioritising information that is consistent with their beliefs, values, motivations and interests [17]. The three key elements of the SCF are: social constructions, target populations and power relations. Social constructions refer to political or policy elements connected with the perceptions and symbols associated with target populations [16,18]. Target populations are defined as those affected by a policy design; that is being the ‘target’ of decision makers. Target populations are normally chosen with the intention of changing their behaviour through the incentives or sanctions attached to the selected policies [19]. For AMR this may be front line health workers and their prescription practices, informal providers of antibiotics or farmers with livestock. Schneider and Ingram consider target populations as key components in the policy process and instrumental in building support and/or opposition to policy design and implementation [18]. Finally, power is defined in terms of three related dimensions: power through decision making; power through control of the agenda; and power through domination [20]. While this approach has its merits, alternative conceptualisations of power, such as Bourdieu’s (1986) and others may have much to offer in studying AMR policy design [21].

From a Social Construction perspective, policies are constructed drawing on symbolic interpretations rather than on objective assumptions [18]. Within this framework, distinct social constructions can lead to different policy options. In the context of selecting between policy options when forming a national action plan on AMR, the SCF-based approach views alternative policies in terms of the benefits or burdens to certain groups, which are associated with positive or negative social constructions. Support for or opposition to policy options is also influenced by the extent to which target populations have political power. These considerations by policymakers, among others, ultimately serve to prioritise or deprioritise specific AMR mitigation policies [18].

## 4. Operationalising the Governance and Social Construction Framework

We have operationalised this new framework vis-a-vis policy setting on appropriate antimicrobial use in Pakistan and are in the process of applying it in Cambodia, Tanzania and Indonesia (Table 1). 

There were two main stages in our Pakistan study. First, we undertook a comprehensive mapping of the range of policy actors involved in policy processes relating to appropriate use of antimicrobials, across the One Health spectrum, including private and public sector actors, using a snowballing process. During this process we asked informants to identify people or organisations that are shaping the approach taken by the government to reduce inappropriate use of antibiotics in human and animal populations in the country. The aim of this mapping exercise is to understand fully the range of policy actors that influence policy around appropriate use of antimicrobials and the linkages between them. During the second stage, between three and six of the most important policy responses (or broad categories of policy responses) were identified from systematic literature reviews, and adapted to the country context. These ‘hypothetical policy options’ are then tested with policy actors identified during the mapping exercise through in-depth interviews featuring a card-sorting game in which interviewees are encouraged to talk through their rationale for placing cards based on a ‘thinking aloud’ methodological technique [22]. Box 1 illustrates the methods applied successfully during 48 interviews in Pakistan. In this study the three broad options used were encourage (the softest option), constrain (restriction rather than ban) and prohibit (the hardest option) [23]. During the interviews we asked policy actors their views on which of these three types of interventions they would support to address the actions of specific antibiotic suppliers. We also asked them to engage in a card-sorting game (Box 1). Through this interview process we were able to solicit new information about preferred policy options; the contextual factors influencing these; the key motivations as described by policy makers; the allocation of benefits and burdens to target groups by public policy; their positive or negative construction; power relations that inform policy choices and ultimately who can potentially benefit from policy change. In Cambodia we used more specific policy implementation strategies for policy actors to respond to in the card sorting exercise; these include restricting doctors to selling only emergency medicines (excluding antibiotics), requiring all pharmacies to have an appropriately trained healthcare provider available for patient consultations at all times, and prohibiting of antibiotic sales without a doctor’s prescription.

This study was a rich source of both quantitative and qualitative data. Some interviewees became so involved with the card-sorting task that they appeared to forget they were being recorded, and so were possibly more open about their views on recommended interventions than they would otherwise have been. In contrast, some interviewees, particularly those involved in implementing regulations, stated they were reticent to speak freely while being recorded. 

While the task acted as a catalyst for some interviewees others did not complete it as it was relatively time-consuming. Some interviewees, for example politicians, did not have time to do the task during the interview and may have felt uncomfortable being asked to do something that they judged inappropriate to someone of their status.

Box 1Description of card-sorting exercise used during interviews in Pakistan.Three sheets of paper labelled Encourage, Constrain or Prohibit were arranged in front of the interviewee, while the difference between interventions that would fall into these three categories was explained using relevant examples. The interviewee was then shown cards representing 23 actors and asked to indicate what type of interventions they proposed to reduce inappropriate use of antibiotics by that actor. A mixture of responses (for example, encourage and constrain) was acceptable. Interviewees were also given the option of saying that they did not have sufficient knowledge to comment on a particular actor, to omit actor cards if they preferred, and to state that they did not consider an actor relevant for the exercise (with explanation).

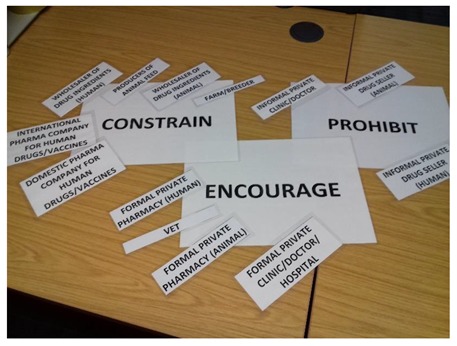


## 5. Conclusions

Progress in policy responses for combating AMR is being hindered, in part, owing to an insufficient evidence base for the effectiveness of the myriad policies proposed targeting in turn the vast array of actors involved across the human health and animal sectors. Although substantial resources are being sought towards assessing the effectiveness of different interventions to combat AMR in LMICs, it would be impossible—and unwise in our opinion—to try and evaluate all interventions in a range of LMIC contexts. Decisions therefore need to be made about which interventions to focus on evaluating first. 

We propose that an understanding of the diverse set of actors (including target populations) that will influence policymakers in LMICs is critical, as well as consideration of the political factors, the context and considerations likely to shape their views, and determine their opposition and support for alternative policy options. Our discussions of the frameworks, methods and the application of policy analysis theories allows for the production of a detailed analysis of how policy actors with conflicting interests negotiate policy and how power relations and networks shape these interactions. From a social construction paradigm we also take into account differences in social constructions and political power with the underlying premise that social constructions can emerge from subjective emotional reactions and are infrequently based on evidence-based practice. An added contribution of our proposed policy analysis in the context of responses to AMR using a One Health approach, is to recognise that actors influencing policy may be particularly diverse and hidden and may not connect to each other, especially across sectors. 

Whilst the SCF can provide a deeper understanding of the policy and politics of AMR, it is also important to mention that there are challenges in operationalizing a more empirically informed approach such as ‘governance’ within a critical theory paradigm. It requires a deep understanding of distinct sociological theories of knowledge and a much more complex data collection process than that required in a standard policy analysis. Equally, this is a theoretical framework developed to be tested and the authors plan to build, expand and adapt this based on knowledge generated by applying it in diverse settings. We thus encourage further theoretical and methodological advances in the field of health policy and cross-sectoral systems research in LMICs, and believe that such studies are critical to tackle the wicked problem AMR represents.

## Figures and Tables

**Table 1 antibiotics-08-00003-t001:** Application of the new framework in diverse low- and middle-income country (LMIC) settings.

Country	Study Focus	Stage
Pakistan	Policy formulation and implementation to reduce inappropriate antimicrobial use in human and animal populations	Study completed December 2018
Cambodia	Policy implementation to reduce inappropriate use of antibiotics through unregistered (human) drug sellers	Study initiated February 2018; to be completed by September 2019
Indonesia	Addressing access to antibiotics without prescription from private drug sellers	Study to be initiated in early 2019
Tanzania	Policy analysis of drivers of antimicrobial resistance across the One Health spectrum in Tanzania	Study funded in November 2018

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
