# Peer review of "Something Borrowed, Something New: A Governance and Social Construction Framework to Investigate Power Relations and Responses of Diverse Stakeholders to Policies Addressing Antimicrobial Resistance"

_antibiotics, 2018, doi:10.3390/antibiotics8010003_

Round 1
Reviewer 1 Report
AUTHORS
Manuscript ID: antibiotics-412413
Title: Something borrowed, something new: a governance and social construction framework to investigate responses of diverse stakeholders to policies addressing antimicrobial resistance
The present study is a short (but meaningful) paper on the role of policy analysis focusing on social constructions, governance and power relations in soliciting a better understanding of support and opposition by key stakeholders for alternative antimicrobial resistance (AMR) mitigation policies. There is very few things (if any) to be said negatively and as such I advise the paper for publication as it is.
Author Response
We would like to thank you for your positive comments on our manuscript, and for advising it to be published as is.
Reviewer 2 Report
In my opinion, the brief manuscript by Helena Legido-Quigley and colleagues is clear and well-written, provides sufficient background information, and limitations are adequately acknowledged. I have no major comments.
Minor comments
Although this is a concept/narrative paper and not a systematic assessment of the literature, I think the manuscript would benefit from a brief method section detailing literature searches and references selection (e.g., structured/unstructured search, which scientific/non-scientific databases were used, how references ultimately included in the paper were selected [e.g., only those fulfilling some predefined criteria or if conversely the selection was based upon subjective authors’ experience or impression]).
Author Response
We would like to thank Reviewer 2 for their positive comments on our paper, and their insightful suggestion. We agree that the addition of a brief methods section would make the paper stronger. We have added the text below to the paper:
To develop our conceptual framework we relied on seminal papers in three areas: governance, social construction and the policy process in LMICs (8-21). Two of the authors reviewed this literature and identified key concepts on governance and how AMR and target populations of potential interventions are socially constructed, while the other two authors operationalised the conceptual framework into a methodology to apply in a range of countries. In the sections that follow, we summarise how the literature we reviewed shaped our conceptual framework and describe the processes used to operationalise the framework in a range of countries.
